# Deep and Wide Transfer Learning with Kernel Matching for Pooling Data from Electroencephalography and Psychological Questionnaires

**DOI:** 10.3390/s21155105

**Published:** 2021-07-28

**Authors:** Diego Fabian Collazos-Huertas, Luisa Fernanda Velasquez-Martinez, Hernan Dario Perez-Nastar, Andres Marino Alvarez-Meza, German Castellanos-Dominguez

**Affiliations:** Signal Processing and Recognition Group, Universidad Nacional de Colombia, Manizales 170001, Colombia; lfvelasquezm@unal.edu.co (L.F.V.-M.); hdperezn@unal.edu.co (H.D.P.-N.); amalvarezme@unal.edu.co (A.M.A.-M.); cgcastellanosd@unal.edu.co (G.C.-D.)

**Keywords:** kernel-embedding, transfer learning, Deep and Wide network, motor imagery

## Abstract

Motor imagery (MI) promotes motor learning and encourages brain–computer interface systems that entail electroencephalogram (EEG) decoding. However, a long period of training is required to master brain rhythms’ self-regulation, resulting in users with MI inefficiency. We introduce a parameter-based approach of cross-subject transfer-learning to improve the performances of poor-performing individuals in MI-based BCI systems, pooling data from labeled EEG measurements and psychological questionnaires via kernel-embedding. To this end, a Deep and Wide neural network for MI classification is implemented to pre-train the network from the source domain. Then, the parameter layers are transferred to initialize the target network within a fine-tuning procedure to recompute the Multilayer Perceptron-based accuracy. To perform data-fusion combining categorical features with the real-valued features, we implement stepwise kernel-matching via Gaussian-embedding. Finally, the paired source–target sets are selected for evaluation purposes according to the inefficiency-based clustering by subjects to consider their influence on BCI motor skills, exploring two choosing strategies of the best-performing subjects (source space): single-subject and multiple-subjects. Validation results achieved for discriminant MI tasks demonstrate that the introduced Deep and Wide neural network presents competitive performance of accuracy even after the inclusion of questionnaire data.

## 1. Introduction

Motor imagery (MI) is related to the process of mentally generating a quasi-perceptual experience in the absence of any appropriate external stimuli [1]. MI practice promotes children’s motor learning and has been suggested to provide benefits in enhancing the musicality of untrained children [2,3], in evaluating the screen-time and cognitive development [4], and improving attentional focus and rehabilitation [5,6,7], among others. MI-based brain–computer interface (BCI) systems often entail electroencephalogram (EEG)-decoding because of their ease of use, safety, high portability, relatively low cost, and, most importantly, high temporal resolution [8]. EEG is a non-invasive and portable neuroimaging technique that records brain electrical signals over the scalp, reflecting the synchronized oscillatory activity originating from the pyramidal cells of the sensorimotor cortex. However, evoked responses in frequency bands, besides the eliciting stimuli, depend upon every individual. In addition, in MI-based cognitive tasks, the evoked event-related de/synchronization of the sensorimotor area is perturbed by other background brain processes or even artifacts, seriously reducing the signal-to-noise ratio [9]. Hence, to generate steady evoked control patterns, long training must master brain rhythms’ self-regulation. As a result, the percentage of users with MI inefficiency (or BCI-illiteracy) is high enough to limit this technology to lab environments even that MI research has been going for many years [10].

In practice, the MI ability can be assessed to determine to what extent a user engages in a mental representation of movements, mainly through self-report questionnaires developed explicitly for this purpose [11]. Yet, there is very little evidence stating a confident correlation between the classification accuracy and the questionnaire scores. Several reasons may account in this regard [12,13]: weak and ambiguous self-interpretation in understanding the questionnaire instructions, laboratory paradigms restricted to a narrow class of motor activity, timeline limitations guaranteeing consistent mental states, and difficulty in learning features from subjects with BCI-illiteracy, among others. Hence, although psychological assessment and questionnaires are probably the most accepted and validated methods in medical contexts [14], their inclusion in the automated prediction of the BCI skills remains very rare due to their disputed reliability and reproducibility [15]. For enhancing the predictive utility, the joint analysis of different imaging modalities is achieved, which may explain the discovered relationships between anatomical, functional, and electrophysiological properties of the brain [16,17]. Nonetheless, besides those issues that may arise by the questionary implementation, research endeavors of multimodal analysis pose a challenging problem in terms of combining categorical data with imaging measurements, facing the following restrictions [18,19]: Different spatial and temporal sampling rates, noninstantaneous and nonlinear coupling, low signal-to-noise ratios, a lack of interpretable results, and the optimal combination of individual modalities is still undetermined, as well as effective dimensionality reduction to enhance the discriminability of extracted multi-view features [20].

Another approach to improve BCI skills is to perform several training sessions in which participants learn how to modulate their sensorimotor rhythms appropriately, relying on the spatial specificity of MI-induced brain plasticity [21]. However, collecting extensive data is time-consuming and mentally exhausting during a prolonged recording session, deteriorating the measurement quality. To overcome this lack of subject-specific data, transfer learning-based approaches are increasingly integrated into MI systems using pre-existing information from other subjects (source domain) to facilitate the calibration for a new subject (target domain) through a set of shared features among individuals under the assumption of a unique data acquisition paradigm [22,23,24]. Therefore, to have the advantages of transfer learning in EEG signal analysis, strategies for individual difference matching and data requirement reduction are needed to fine-tune the model for the target subject [25]. For example, in [26], the authors use pre-trained models (e.g., VGG16 and Alex-net) as the starting point for approach-fitting. This strategy limits the amount of training data required to support the MI classification task. In this case, they compute the continuous wavelet transform from EEG signals to represent the time-series data into equivalent image representation that can be trained in deep networks. Similarly, Zhang et al. in [27] proposed five schemes for adaptation of a deep convolutional neural network-based EEG-BCI system for decoding MI. Specifically, each procedure fine-tunes a pre-trained model to enhance the evaluation performed on a target subject. Recently, approaches based on weighted instances [28] and domain adaptation [29] have been studied. In the first case, instance-based transfer learning is used to select the source domain data that is most similar to the target domain to assist the training of the target domain classification model. In the second case, researchers extend deep transfer learning techniques to the EEG multi-subject training case. In particular, they explore the possibility of applying maximum-mean discrepancy to align better distributions of features from individual feature extractors in an MI-based BCI system. Nonetheless, to extract sets of shared features among subjects with a similar distribution, there is a need to adequately handle two main limitations of subject-dependent and subject-independent training strategies: small-scale datasets and a significant difference in signals across subjects [30]. In fact, several issues remain as challenges to obtaining adequate consistency of the feature space and probability distribution of training and test data, avoiding negative transfer effects [31,32]: feature extraction from available multimodal data effective enough to discriminate between MI tasks, and the choosing of transferable objects and transferability measures along with the assignation of their weights [33].

Here, we introduce a parameter-based approach of cross-subject transfer learning for improving poor-performing individuals in MI-based BCI systems, and pooling data from labeled EEG measurements and psychological questionnaires via kernel-embedding. For sharing the discovered model parameters, as presented in [34], an end-to-end Deep and Wide neural network for MI classification is implemented that is, firstly, fed by data from the whole trial set to pre-train the network from the source domain. Then, the layer parameter layers are transferred to initialize the target network within a fine-tuning procedure to recompute the Multilayer Perceptron-based accuracy. To perform data fusion combining categoricals with the real-valued features, we implement the stepwise kernel-matching via Gaussian embedding, resulting in similarity matrices that hold a relationship with the BCI inefficiency clusters. For evaluation purposes, the paired source–target sets are selected according to the inefficiency-based clustering by subjects to consider their influence on BCI motor skills, exploring two choosing strategies of the best-performing subjects (source space): Single-subject and multiple-subjects, as delivered in [35]. The validation results for discriminating MI tasks show that the proposed Deep and Wide neural network gives promising accuracy performance, even after including questionnaire data. Therefore, this deep learning framework with cross-subject transfer learning is a promising way to address small-scale data limitations from the best-performing subjects.

The remainder of this paper is as follows: Section 2 presents the materials and methods, Section 3 describes the experiments and the corresponding results, putting effort into their interpretation. Lastly, Section 4 highlights the conclusions and recommendations.

## 2. Materials and Methods

### 2.1. 2D Feature Representation of EEG Data

From the EEG database collected by an *C*-channel montage, we build a single matrix for the *n*-th trial ${\left\{X_{n}\in R^{C\;\times \;T},\lambda_{n}\in \;{\left\{0,1\right\}}^{\Lambda }\right\}}_{n=1}^{N},$ that contains *T* time points at the sampling rate $F_{s}$. Along with the EEG data, we also create the one-hot output vector $\lambda_{n}$ in Λ∈N labels. For evaluation in discriminating MI tasks, the proposed transfer learning model is assessed on a trial basis. That is, we extract the feature sets per trial ${\left\{{\hat{X}}_{n}^{r}\in R^{C}\right\}}_{r=1}^{R}$, incorporating a pair of EEG-based feature representation approaches (R=2): Continuous Wavelet Transform (CWT) and Common Spatial Patterns (CSP), as recommended for Deep and Wide learning frameworks in [36].

Further, the extracted multi-channel features (using CSP and CWT methods) are converted into a two-dimensional topographic interpolation $R^{C}\;\;\to \;\;R^{W\;\times \;H}$ to preserve their spatial interpretation, mapping into a two-dimensional circular view for every extracted trial feature set. As a result, we obtain the labeled 2D data $\left\{Y_{n}^{z}\in R^{W\;\times \;H},\lambda_{n}\;\;:\;\;n\in N\right\}$, where $Y_{n}^{z}$ is a single-trial bi-domain *t-f* feature array, termed *topogram*, extracted from every *z*-th set. Of note, the triplet $z=\;\{r,\Delta_{t},\Delta_{f}$ (with z∈Z) indexes a topogram estimated for each included domain principle r∈R at the time-segment Δt∈T, and within the frequency-band Δf∈F.

Besides, we estimate the local spatial patterns of relationships from the input topographic set through the square-shaped layer kernel arrangement ${\left\{K_{i,l}^{z}\in R^{P\;\times \;P}\right\}}^{I_{l},Z}$ (as in straightforward convolutional networks), where *P* holds the kernel size. Therefore, the number of kernels varies at each layer $i\in I_{l}$, so that the stepwise 2D-convolutional operation is performed over the input topogram, $Y^{z}$, as follows: (1)$${\hat{Y}}_{L}^{z}=\left(\varphi_{L}^{z}\circ \cdots \circ \varphi_{1}^{z}\right)\left(Y^{z}\right),$$
where $\varphi_{l}^{z}\left({\hat{Y}}_{l-1}^{z}\right)=\gamma_{l}\left(K_{i,l}^{z}⊗{\hat{Y}}_{l-1}^{z}+B_{i,l}^{z}\right)$ is the convolutional layer, followed by a non-linear activation function $\gamma_{l}\;\;:\;\;R^{W_{l}^{z}\;\times \;H_{l}^{z}}\;\;\to \;\;R^{W_{l}^{z}\;\times \;H_{l}^{z}}$, ${\hat{Y}}_{l}^{z}\in R^{W_{l}^{z}\;\times \;H_{l}^{z}}$ is the resulting 2D feature map of the *l*-th layer (adjusting ${\hat{Y}}_{0}^{z}=Y^{z}$), and the arrangement $B_{i,l}^{z}\in R^{W_{l}^{z}\;\times \;H_{l}^{z}}$ denotes the bias matrix. Notations ∘ and ⊗ stand, respectively, for the function composition and convolution operator.

### 2.2. Multi-Layer Perceptron Classifier Using 2D Feature Representation

In this stage, we employ the deep learning-based classifier function $\varphi \;\;:\;\;R^{W\;\times \;H}\mapsto \Lambda$ developed through a Multilayer Perceptron (MLP) Neural Network that predicts the label probability vector $\tilde{v}\in {\left\{0,1\right\}}^{\Lambda }$, as below [37]:
(2a)$$\begin{array}{cc} \tilde{v} & =\varphi \left(u_{0},\Theta ;\phi_{D}^{z}\circ \cdots \circ \phi_{1}^{z}\right), \end{array}$$
(2b)$$\begin{array}{cc} s.t.:\; & \Theta_{0}^{*}=\arg\min_{K_{i,l}^{z},A_{d},B_{i,l}^{z},\alpha_{d}}\left\{L\left({\tilde{v}}_{n},\lambda_{n}|\Theta \right);\;\forall n\in N\right\} \end{array}$$
where $\phi_{d}\left(u_{d-1}\right)=\eta_{d}\left(A_{d}u_{d-1}+\alpha_{d}\right)$ is the fully-connected layer ruled by the non-linear activation function: $\eta_{d}\;\;:\;\;R^{P_{d}^{\prime }}\;\;\to \;\;R^{P_{d}^{\prime }}$, $P_{d}^{\prime }\in N$ is the number of hidden units at the *d*-th layer, d={0,…,D} (d=0 is the initial concatenation before the classification layer), $A_{d}\in R^{P_{d}^{\prime }\;\times \;P_{d-1}^{\prime }}$ is the weighting matrix containing the connection weights between the preceding neurons and the hidden units $P^{\prime }$ of layer *d*, $\alpha_{d}\in R^{P_{d}^{\prime }}$ is the bias vector, and the $u_{d}\in R^{P_{d}^{\prime }}$ hidden layer vector holds the extracted spatial information encoded by the resulting 2D feature maps in the *Q* domain.

For computation at each layer, the hidden layer vector is iteratively updated by the rule (composition function-based approach of deep learning methods) $u_{d}=\phi_{d}\left(u_{d-1}\right)$, for which the initial state vector is flattened by concatenating all matrix rows across *z* and $I_{l}$ domains as $u_{0}=\;\left[\mathrm{vec}\left({\hat{Y}}_{L}^{z}\right)\;\;:\;\;\forall z\in Z\right]$. The input vector $u_{0}$ sizes $G=W^{\prime }H^{\prime }Z\sum_{l\in L}I_{l}$, holding $W^{\prime }\;\;<\;\;W,H\;\;<\;\;H^{\prime }$. Besides, the optimizing estimation framework of label adjustment estimates the training parameter set $\Theta_{0}=\;\left\{K_{i,l}^{z},A_{d},b_{i,l}^{z},\alpha_{d}\right\}$, fixing the loss function $L\;\;:\;\;R^{\Lambda }\times R^{\Lambda }\;\;\to \;\;R$ to calculate the gradients employed to update the weights and bias of the proposed Deep and Wide neural network through a certain number of training epochs. Remarkably, we refer to our method as Deep and Wide because of the inclusion of a set of different topograms (along time and frequency domains) from the extracted multi-channel features using CSP and CWT algorithms. A mini-batch-based gradient implements the solution, as commonly used in deep learning methods, equipped with automatic differentiation and back-propagation [38].

### 2.3. Transfer Learning with Added Questionnaire Data

In EEG analysis based on Deep Learning, for enhancing the classifier performance, transfer learning is a common approach to adjust a pre-trained neural network model equipped with the label probability vector $\tilde{v}$, aiming to provide a close domain distance measurement $\delta \left(\cdot ,\cdot \right)R^{+}$, lower than a given value $\epsilon \in R^{+}$, between the paired domains to approximate the source $Y^{\left(s\right)}$ to the target $Y^{\left(t\right)}$ [24], as follows: (3)$$\begin{array}{cc} \delta (Y^{\left(s\right)}\left(Y,S\right),Y^{\left(t\right)}\left(Y,S\right)|\tilde{v}) & \leq \epsilon \end{array}$$(4)$$\begin{array}{c} s.t.:\;\Theta^{*}=\left\{K_{i,l}^{q*},B_{i,l}^{q*}\right\} \end{array}$$

Here, we propose to conduct the transfer learning procedure to learn a target prediction function that is enhanced by the addition of the categorical assessments of a psychological questionnaire data matrix, S, along with the stepwise multi-space kernel-embedding, including EEG-based features, to perform the whole network parameter optimization in Equation (2b). Besides, for interpretation purposes, selecting the paired source–target sets is accomplished according to the inefficiency-based clustering of subjects.

Therefore, to combine the categorical data, S, with the real-valued feature map set extracted from EEG as exposed in Section 2.1 and Section 2.2, Y, we compute the tensor product space between the corresponding kernel-matching representations, $\kappa_{\hat{U}}$ and $\kappa_{S}$, as suggested in [39]: (5)$$\bar{\kappa }=\kappa_{\hat{U}}\circ \kappa_{S},\;\bar{\kappa }\in R^{J\;\times \;J}$$
where $J=\sum_{m=1}^{M}N_{m}$ ($N_{m}$ holds the trials for the *m*-th subject), $\kappa_{S}\in R^{J\;\times \;J}$ is the kernel matrix directly extracted from the questionnaire data $S\in R^{J\;\times \;N_{Q}}$ ($N_{Q}$ is the questionnaire vector length), $\kappa_{\hat{u}}\in R^{J\;\times \;J}$ is the kernel topographic matrix estimated from the projected version $\hat{U}=U\Upsilon^{*}$, with $\hat{U}\in R^{J\;\times \;G^{\prime }}$ (holding that $G^{\prime }<G$), $U\in R^{J\;\times \;G}$$U\in R^{J\;\times \;G}$ is the initial data matrix build by concatenating across the trial and subject sets all flattened vectors $u_{0}^{*}$, which are computed by adjusting the optimized parameters $\Theta^{*}=\left\{K_{i,l}^{q*},B_{i,l}^{q*}\right\}$, and $\Upsilon^{*}\in R^{G\;\times \;G^{\prime }}$ is the projection matrix introduced to maximize the similarity between both estimated kernel-embeddings derived from the labeled EEG measurements of MI responses, namely, one from the one-hot label vectors, $\kappa_{V}\in R^{J\;\times \;J}$, and another from the topographic features, $\kappa_{U}\in R^{J\;\times \;J}$.

In particular, we match both estimated kernel-embeddings through the centered kernel alignment (CKA), as detailed in [40]: (6)$$\Upsilon^{*}=\arg\max_{\Upsilon }\mathrm{CKA}\left(\kappa_{U},\kappa_{V}\right)$$
where the kernel $\kappa_{V}$ is obtained from the matrix of predicted label probabilities $V\in R^{J\;\times \;\Lambda }$ build by concatenating across the trial and subject sets all label probability vectors ${\tilde{v}}_{mn}$.

## 3. Experimental Set-Up

Training of the proposed Deep and Wide neural network model for transfer learning to improve classification of MI responses, including EEG and questionnaire data, encompasses the following stages (see Figure 1): (i) Preprocessing and spatial filtering of EEG signals, followed by 2D features extracted from the input topogram set using the convolutional network (see Section 2.1). (ii) MLP classification applying the extracted 2D feature maps (see Section 2.2), (iii) Cross-subject transfer learning, including stepwise multi-space kernel-embedding of the real-valued and categorical variables (see Section 2.3). The paired source–target sets are selected according to the inefficiency-based clustering by subjects to consider their influence on BCI motor skills.

Nonetheless, the classifier performance can decrease since the extracted representation sets may still involve irrelevant and/or similar features. Therefore, for reducing the data complexity, we accomplish dimensionality reduction by evaluating a widely-used unsupervised feature extractor of Kernel PCA (KPCA) that provides a representation of data points’ global structure [41].

### 3.1. Database Description and Preprocessing

*GigaScience* (publicly available at http://gigadb.org/dataset/100295 (accessed on 9 July 2021)): This acquisition holds EEG data recorded by a BCI experimental paradigm of MI movement collected from 52 subjects (though only 50 is available). Data were acquired by a 10–10 *C*-electrode system C=64 with 512 Hz sampling rates, collecting 100 individual trials (each one lasting 7 s) in either task (left or right hand). The MI paradigm begins with a fixation cross presented on a black screen within 2 s. Next, a cue instruction appeared randomly on the screen for 3 s to ask each subject to imagine moving the fingers, starting to from the forefinger and proceeding to the little finger, touching each to their thumb. A blank screen was then shown at the beginning of a break period, lasting randomly between 4.1 and 4.8 s. For each MI class, these procedures were repeated 20 times within a single testing run.

*GigaScience* also collected subjective answers to physiological and psychological questionnaires (categorical data), intending to investigate the evidence on performance variations to work out strategies of subject-to-subject transfer in response to intersubject variability. To this end, all subjects were invited to fill out a questionnaire during three different phases of the MI paradigm timeline: before beginning the experiment (each subject answered $N_{Q}=15$ questions); after every run within the experiment ($N_{Q}=10$ questions were answered); and at the experiment’s termination ($N_{Q}=4$ answered questions, $\left\{Q_{i}\;\;:\;\;i=1,2,3,4\right\}$).

As preprocessing, we filtered each raw channel $x_{n}^{c}\in R^{T}$ within [8–30] Hz using a five-order Butterworth band-pass filter. Further, we carry out a bi-domain short-time feature extraction (i.e., CWT and CSP—see Section 2.1), as performed in [42]. In the former extraction, the wavelet coefficients are assumed to provide a compact representation pinpointing the EEG data energy distribution, yielding a time-frequency map in which the amplitudes of individual frequencies (rather than frequency bands) are represented. In the latter extraction, the goal of CSP is to employ a linear relationship to transfer a multi-channel EEG dataset into a subspace with a lower dimension (i.e., latent source space), aiming to enhance the class separability by maximizing the labeled covariance in the latent space. In both extraction cases, we fix the sliding short-time window length parameter $\tau \in R^{+}$ according to the accuracy achieved by the baseline Filter Bank CSP algorithm that is performed using the whole range of considered frequency bands. The sliding window is adjusted to τ=2 s with a step size of 1 s as an appropriate choice to extract $N_{\tau }=5$ EEG segments, as performed in [43]. Since electrical brain activities provoked by MI tasks are commonly related to μ and β rhythms [44], the spectral range is split into the following bandwidths of interest: Δf∈{μ∈[8–12],β∈[12–30]} Hz. The CWT feature set is computed by the Complex Morlet function frequently applied in the spectral EEG analysis, fixing a scaling value to 32. Additionally, we set the number of CSP components as 3Λ (Λ∈N holds the number of MI tasks), utilizing a regularized sample covariance estimation.

### 3.2. MLP Classifier Performance Fed by 2D Features

At this stage, we carry out the extraction of 2D feature maps from the input topogram set using the convolutional network. Further, the 2D features extracted to feed the MLP-based classifier with the parameter tuning shown in Table 1, and the resulting layer-by-layer model architecture is illustrated in Figure 2. For implementation purposes, we apply the Adam algorithm using the optimizing procedure with fixed parameters: a learning rate of $1\times 10^{-3}$, 200 training epochs, and a batch size of 256 samples. Additionally, the mean squared error (MSE) is chosen as the loss function L(:) in Equation (2b), that is, $L\left({\tilde{v}}_{n},\lambda_{n}|\Theta \right)\;=\;E\left\{{\left({\tilde{v}}_{n}-\lambda_{n}\right)}^{2}\right\}$. For speeding the learning procedure, the Deep and Wide neural network framework is written in Python code (TensorFlow toolbox and Keras API) trained to employ multiple GPU devices at the Google Colaboratory. The codes are made available at a public GitHub repository (codes available at https://github.com/dfcollazosh/DWCNN_TL (accessed on 9 July 2021)).

As the performance measure, the classifier accuracy $A_{c}\in \left[0,1\right]$ is computed by the expression: $A_{c}=\;\left(T_{P}\;\;+\;\;T_{N}\right)/\left(T_{P}\;\;+\;\;T_{N}\;\;+\;\;F_{P}\;\;+\;\;F_{N}\right)$, where $T_{P}$, $T_{N}$, $F_{P}$, and $F_{N}$ are true-positives, true-negatives, false-positives, and false-negatives, respectively. In this case, we split the subject’s dataset and built the training set using 90% of trials and the remaining 10% for the test set. Further, the individual training trial set is randomly partitioned by a stratified 10-fold cross-validation to generate a validation trial partition.

For the tested subject set, Figure 3 displays the results of accuracy that the MLP-based classifier produces if fed by just the 2D feature set extracted before. From the obtained accuracy values, we evaluate the performance to be considered as inadequate in brain–computer interface systems as detailed in [45]. Namely, we cluster the individual set into the following three groups with distinctive BCI skills: (i) Group of individuals performing the highest accuracy but with very low variability of neural responses (colored in green). (ii) A group that reaches superior classifier performance but with some response fluctuations (yellow color). (iii) A group that produces modest performance along with a high unevenness of responses (red color).

### 3.3. Performed Stepwise Multi-Space Kernel Matching

Algorithm 1 presents the procedures to complete the validation of the suggested transfer learning with multi-space kernel-embedding. We implement the Gaussian kernel to represent the available data because of its universal approximating ability and mathematical tractability. The length scale hyperparameter $\sigma \in R^{+}$, ruling the variance of the described data, is adjusted to their median estimate. The following steps (3: and 4:) accomplish the pairwise kernel matching, firstly between the sets of EEG measurement U and label probability V. To this end, the CKA matching estimator is fed by the concatenated EEG features together with the predicted label probabilities to perform alignment across the whole subject set, empirically fixing the parameter $G^{\prime }$ to 50 according to the subjects’ number in this experiment. In the second matching, we encode all the available categorical information about the psychological and physiological evaluation with the relevant feature set, resulting from CKA, by their projection onto a common matrix space representation, using the kernel/tensor product. Note that the projected data $\hat{u}$ by CKA are also embedded. We also perform dimensionality reduction of the feature sets generated after stepwise-matching using Kernel Principal Component Analysis (KPCA) for evaluating the representational ability.

Further, we estimate the subject similarity matrix from the extracted feature sets, aiming to assess the domain distance between the source-target pairs, which are to be selected from different clusters of BCI inefficiency. Since the clustering of individuals relies on the ordered accuracy vector, we introduce the following neighboring similarity matrix ${\bar{\Delta }}_{\hat{\xi }}$ with pairwise metric elements computed from the matrices $\hat{\xi }=\left\{\hat{\bar{\kappa }},{\hat{\kappa }}_{KPCA}\right\}$, as follows: (7)$$\begin{array}{cc} {\bar{\Delta }}_{\hat{\xi }}\left(m,m^{\prime }\right) & =\mathrm{cov}\left(\mathrm{seq}\left(\Delta_{\hat{\xi }}\left(m,\forall m^{\prime }\right)\right),\mathrm{seq}\left(\Delta_{\hat{\xi }}\left(m^{\prime },\forall m\right)\right)\right),\;\;{\bar{\Delta }}_{\hat{\xi }}\left(m,m^{\prime }\right)\in {\bar{\Delta }}_{\hat{\xi }}\in R^{M\;\times \;M}\\ \Delta_{\hat{\xi }}\left(m,m^{\prime }\right) & =\sum_{\forall j\in J}{\left|\hat{\xi }\left(m,j\right)-\hat{\xi }\left(m^{\prime },j\right)\right|}^{2},\;\Delta_{\hat{\xi }}\left(m,m^{\prime }\right)\in \Delta_{\hat{\xi }}\in R^{M\;\times \;M} \end{array}$$
where notations cov(·,·) and $\mathrm{seq}\left(\Delta \left(m,\forall m^{\prime }\right)\right)\in R^{M}$ stand for, respectively, the covariance operator and the sequence composed of all $\forall m^{\prime }\in M$ elements of row *m* ranked in decreasing order of the achieved MLP-based accuracy. The rationale for applying the covariance over the ranked row vectors of $\Delta_{\hat{\xi }}$ is to preserve the similarity information between neighboring subjects.
**Algorithm 1** Validation procedure of the proposed approach for transfer learning with stepwise, multi-space kernel matching. ${}^{†}$ Dimensionality reduction is an optional procedure performed for comparison purposes.**Input data**: EEG measurement U, predicted label probabilities V, questionary data S,∀m∈M1:Initial parameter set estimation$\Theta_{0}^{*}$: Compute the baseline MLP-based accuracy from U and V by optimizing $\Theta =\{K_{i,l}^{q},A_{d},B_{i,l}^{q},\alpha_{d}\}$2:**for**∀m∈M, n∈N**do**3:    Kernel matching between EEG measurement U and labels V,    Compute kernel-embedding of input data $\xi =\{\kappa_{U},\kappa_{S},\kappa_{V}\}$: $\kappa_{\xi }=N_{\xi }\left(\mu_{\xi },\sigma_{\xi }\right)$    Compute Center Kernel Alignment between both spaces: $\mathrm{CKA}\left(\kappa_{U},\kappa_{V}\right)$4:    Kernel matching on supervised EEG representation for the categorical data    Compute kernel-embedding of projected data $\hat{U}$ using $\kappa_{\hat{U}}=N_{\hat{U}}\left(\mu_{\hat{U}},\sigma_{\hat{U}}\right)$    Compute tensor product, including the categorical data $\bar{\kappa }=\kappa_{\hat{U}}\circ \kappa_{S},\;\bar{\kappa }\in R^{J\;\times \;J},\;J=NM$5:**end for**Dimensionality reduction ${}^{†}$ by Kernel Principal Components: ${\bar{\kappa }}_{\mathrm{KPCA}}\in R^{J\;\times \;J}$6:Transfer learning of paired source-target subjects: $Y^{\left(s\right)}$ and $Y^{\left(t\right)}$    Perform matrix reshaping $R^{J\;\times \;J}\mapsto R^{M\;\times \;J}$: $\hat{\xi }=\left\{\hat{\bar{\kappa }},{\hat{\kappa }}_{KPCA}\right\}$    Compute the neighboring similarity matrix of individuals: $\bar{\Delta },{\bar{\Delta }}_{KPCA}$    Compute the intra-subject distance matrix through the domain distance measurement: ${\bar{\delta }}_{\hat{\xi }}\left(m\right)\in R^{+},\forall m\in M$    Select paired subjects for each transfer learning strategy evaluated:          (*a*) One-source versus one-target, (*b*) multiple-source versus one-target    Recompute the MLP-based accuracy of targets, initializing the parameter set as $\Theta =\Theta_{0}^{*}$, fixing the $P^{\prime }$ parameter according to the source subject.7:**Output data**: Accuracy gain achieved by each individual target, according to the selection transfer learning strategy evaluated.

Figure 4 displays the similarity matrix performed by the tensor product $\Delta_{\hat{\xi }}$ (left column), evidencing some of the relations between the clustered subjects, but depending on the evaluated questionary data. Thus, the collection $Q_{1}$ yields two groups, while $Q_{4}$ exhibits three partitions. Instead, $Q_{2}$ and $Q_{3}$ do not cluster the individuals precisely. After KPCA dimensionality reduction, however, the proximity assessments ${\bar{\Delta }}_{KPCA}$ tend to make the neighboring association more solid, resulting in clusters of subjects with more distinct feature representation, as shown in the middle column for each questionary.

Under the assumption that the closer the association between the paired source-target couples, the more effectively their cross-subject transfer learning is implemented, we estimate the marginal distance ${\bar{\delta }}_{\hat{\xi }}\left(m\right)\in R^{+}$ from either version $\Delta_{\hat{\xi }},\;{\bar{\Delta }}_{KPCA}$ by averaging the neighboring similarity of each subject over the whole set, as follows: (8)$${\bar{\delta }}_{\hat{\xi }}\left(m\right)=E\left\{|{\bar{\Delta }}_{\hat{\xi }}\left(m,m^{\prime }\right)|:\forall m^{\prime }\in M\right\},$$
where the notation $E\left\{z:\forall \zeta \right\}$ stands for the expectation operator computed across the whole set {ζ}.

The right column displays the values of marginal values ${\bar{\delta }}_{\hat{\xi }}\left(m\right)$, showing that each individual is differently influenced by the stepwise multi-space kernel matching of electroencephalography to psychological questionnaires $Q_{i}$. These results are in agreement with the subject cluster properties evaluated above. Thus, $Q_{1}$ and $Q_{4}$, having more discernible partitions, yield the feature representations that are more even in the subject set, while $Q_{2}$ and $Q_{3}$ provide irregular representations. One more aspect is the effect of dimensionality reduction that improves the representation of $Q_{1}$ and $Q_{2}$ cases. On the contrary, the use of KPCA tends to worsen the global similarity level of individuals.

### 3.4. Estimation of Pre-Trained Weights for Cross-Subject Transfer Learning

The following step is to pair the representation learned on a *source* to be transferred to a given *target* subject. Starting from the subject partitions according to their BCI skills performed above in Section 3.2, we select the candidate sources (i.e., the source space $Y^{\left(s\right)}\left(,\right)$) within the best-performing subjects (Group *I*), while the target space $Y^{\left(t\right)}\left(,\right)$ becomes the worst-performing participants (Group *III*). Here, we validate two choosing strategies of subjects from the source space (Group *I*):(a)*Single source*-*single-target*, when we select the subject of Group *I*, achieving the highest value of the domain distance measurement in Equation (Equation 9) computed as follows:
(9)$$\max_{\forall m\in \mathrm{Group}I}{\bar{\delta }}_{\hat{\xi }}\left(m;Q_{i}\right)$$Once the *source*-*target* pairs are selected, the pre-trained weights are computed from each designed *source* subject to initialize the Deep and Wide neural network, rather than introducing a zero-valued starting iterate, and thus enabling a better convergence of the training algorithm. Note that the fulfilling condition in Equation (Equation 9) depends on $Q_{i}$, meaning distinct selected sources for each questionnaire data.(b)*Multiple sources*-*single-target* when the selected subjects of Group I achieve the four highest domain distance values. In this case, the Deep and Wide initialization procedure applies the pre-trained weights estimated from the concatenation of the source topograms.

Figure 5 details the reached classification performance using the proposed transfer learning approach for either strategy of selecting the candidate sources through a radar diagram that includes all target subjects (axes). For comparison’s sake, the graphical representation depicts, with a line colored in black, the MLP-based accuracy (see Figure 3) as a reference for assessing the performance classifier gain due to the applied transfer learning approach, the accuracy achieved by the features extracted by the tensor product (blue line) and KPCA (magenta line), $\hat{\bar{\kappa }},{\hat{\kappa }}_{KPCA}$, respectively.

The odd columns (first and third) present the *Single source-Single-target* diagrams, while the even ones are for the *Multiple sources-Single-target* strategy. In all cases of questionnaire data $Q_{i}$, the transfer learning with stepwise, multi-space kernel matching allows increasing, on average, the baseline classifier performance of the subjects belonging to Group *III* with modest accuracy and high unevenness of responses. Nevertheless, there are still some issues to be clarified. The accuracy gain performed by the *Single source-Single-target* strategy is lower than the one achieved by the latter approach, but the number of subjects that benefit from the transfer learning approach is higher. On the contrary, the presence of multiple sources halves the number of poor-performing subjects that are improved, though they produce accuracy gain values up to 25% (see subject #45). The next aspect of addressing is the contribution of categorical data in terms of classifier performance. The first two radars in the bottom row (labeled as EEG) present the accuracy improvement performed by the features extracted from the EEG measurements after CKA alignment ($\mathrm{CKA}\left(\kappa_{U},\kappa_{V}\right)$), underperforming the transfer learning adding questionnaires.

Topographic maps of representative subjects (computed with and without transfer learning) using just the feature map information, presenting the learned weights with assumed meaningful activity.

Regarding the dimensional reduction additionally considered, its delivered accuracy (outlined in magenta) strongly depends on the specific case of fused data $Q_{i}$. Thus, while $Q_{1}$ and $Q_{2}$ benefit from the KPCA procedure, $Q_{4}$ reduces the performance achieved. This result becomes evident in two bottom radars (3-th and 4-th) that depict the effect of transfer learning averaged across the data $Q_{i}$, showing that the classifier performance of almost each target individual can be enhanced by the proposed transfer learning approach for either strategy of selecting the candidate sources. However, there are a couple of subjects (# 38 and # 20) that did not have a positive impact.

Lastly, the topographic maps shown in Figure 6 give a visual interpretation of the proposed transfer learning, which are reconstructed from the learned network weights according to the algorithm introduced in [37]. We compare the estimated approaches under the assumption that the discriminating power is directly proportional to the reconstructed weight value. Thus, the top row shows the topograms of the single-source strategy built from both bandwidths (β and μ) within different intervals of the neural response. As seen, the source selected (subject #3) performs a weight set with the spatial distribution related to the sensorimotor area, focusing their neural responses within the MI segment correctly. Next to S #3, we present the target’s topograms that benefit the most from the transfer learning, holding weights with a spatial distribution that is a bit blurred. The effect of the single-source transfer learning approach is the reduction of the weight variability, as shown in the adjacent topograms. However, the source effectiveness to reduce the variability is limited in the case of the low-skilled target #38 that presents many contributing weights spread all over the scalp area. Moreover, the weights appear inside the two intervals (before cue-onset and the ending segment) at which the responses elicited by MI tasks are believed to vanish. As a result, the single-source strategy yields a negative accuracy gain of Target #38 (it drops from 70% to 65%).

Similar behavior is also observed in the second row, displaying the topograms of the multi-source strategy performed by the most benefitting (T#11) and the worst-achieving target (T#22), respectively. However, the inclusion of multiple sources leads to weights with a sparse distribution, as observed in the topograms of the selected subjects (S#3,14,41,28). This effect may explain the small number of targets improved by the multi-source strategy. In order to clarify this point, the bottom row displays the corresponding spatial distribution performed by the multi-source strategy when including the whole subject set of Group I, resulting in weights that are very weak and scattered. Moreover, compared with the first two rows, the all-subjects source approach of the bottom row makes the related transfer learning deliver the worst performance averaged across the target subject set.

## 4. Discussion and Concluding Remarks

Here, we introduce a cross-subject transfer learning approach for improving the classification accuracy of elicited neural responses, pooling data from labeled EEG measurements and psychological questionnaires through a stepwise multi-space kernel-embedding. For validation purposes, the transfer learning is implemented in a Deep and Wide framework, for which the source-target sets are paired according to the BCI inefficiency, showing that the classifier performance of almost each target individual can be enhanced using single or multiple sources.

From the evaluation results, the following aspects are to be highlighted:

*Evaluated NN framework*: The Deep and Wide learning framework is supplied by the 2D feature maps extracted to support the MLP-based classifier. As a result, Table 2 compares the bi-class accuracy of the *GigaScience* database achieved by several recently published approaches, which are outperformed by the learning algorithm with the proposed transfer learning method. Of note, the MSNN algorithm presented in [46] achieves a competitive classification accuracy on average, 82.6 (ours) vs. 81.0 (MSNN), but with a higher standard deviation in comparison with our proposal, 12.0 vs. 8.4. Besides, our method can include categorical data from questionnaires within the MI paradigm, which favors the interpretability concerning the studied subject from spatial, time, and frequency patterns from EEG data coupled with categorical physiological and psychological data.

*Feature representation challenges and computational requirements:* The bi-domain extraction is presented (CWT and CSP) to deal with the substantial intra-subject variability in patterns across trials. However, for improving their combination with categorical data, more compact feature representations can be explored, for instance, using connectivity metrics like in [52]. Besides, neural network architectures capturing the temporal dynamics local structures of the EEG time-series associated with the elicited MI responses could be helpful to upgrade our approach [53]. Moreover, it is well-known that deep learning approaches require considerable computational time when training the model. For clarity, a computational time experiment is carried out. Specifically, for the parameter setting of the FC8 layer, with regularization values l1 and l2 tuned by a grid search around $\left[0.0005,0.001,0.005\right]$, and a number of neurons fixed through a grid search within $P^{\prime }=\left[100,200,300\right]$, the fitting time with and without our transfer learning approach is summarized in Table 3. As seen, the multi-source scheme requires more computation time per fold. Still, real-time BCI requirements can be satisfied once the model is trained, and a new instance must be predicted. In short, for a new subject, the following stages must be carried out: (i) Store the EEG and questionnaire information of the new and training subjects. (ii) Apply our transfer learning approach as exposed in Figure 1 to couple EEG and questionary psychological data for the new subject. (iii) Once the model is trained, new instances of the studied subject can be predicted as straightforward deep learning methods (in this stage, only the EEG data is required).

*Multi-space kernel matching*: To overcome the difficulties in utilizing data-fusion combining categorical with the real-valued features, we implement the stepwise kernel matching via Gaussian embedding. As a consequence, the obtained similarity matrices evidence the relationship with the BCI inefficiency clusters of subjects. Even though the association is highly influenced by each evaluated questionnaire data, this result becomes essential in light of previous reports stating that no statistically significant differences can be detected between questionary scores and EEG-based performance [54]. One more aspect is the effect of dimensionality reduction through kernel PCA that improves the representation, but to a certain extent (only in $Q_{1}$ and $Q_{2}$ cases). For tackling the differences in subjective criteria for predicting MI performance, however, two main issues need to be addressed: The use of more appropriate kernel-embedding for categorical scores [55] and dimensionality reduction approaches, providing representation of data points with a wide range of structures like *t*-Distributed Stochastic Neighbor Embedding [56].

*Cross-subject transfer learning*: We conduct the transfer learning to infer a target prediction function from the kernel spaces embedded before, selecting the paired source-target sets according to the Inefficiency-based clustering by subjects. Overall, the transfer learning with feature representations, combined with questionary data, allows for an increase of the baseline classifier accuracy of the worst-performing subjects. Nevertheless, source selection through a different method impacts the classifier performance; while the *Multiple-source-Single-target* strategy tends to produce accuracy improvements that are bigger than the *Single-source-Single-target*, and the number of the benefited targets declines. This result may point to future exploration of more effective transfer learning of BCI inefficiency devoted to bringing together, as much as possible, the source domain to each target space. This task also implies improving the similarity metric in Equation (Equation 7) proposed for comparing ordered-by-accuracy vectors of different BCI inefficiency clusters.

As future work, the authors plan to validate the cross-subject transfer learning approach in applications with the joint incorporation of two or more databases (cross-database), growing the tested number of individuals significantly. For instance, we plan to consider the dataset collected by the Department of Brain and Cognitive Engineering, Korea University in [57], since this set holds questionnaire data information about the physiological and psychological condition of subjects. As a result, we will obtain classification performances based on transfer learning at intra-subject and inter-dataset levels.

## Figures and Tables

**Figure 1 sensors-21-05105-f001:**
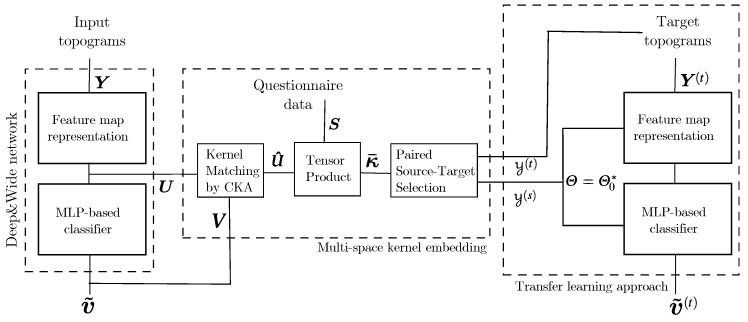
Guideline of the proposed transfer learning approach, including Stepwise Kernel Matching to combine data from Electroencephalography and Psychological Questionnaires.

**Figure 2 sensors-21-05105-f002:**
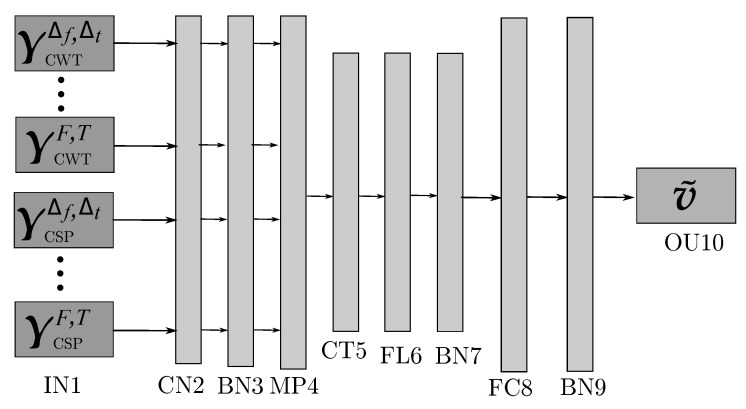
Scheme of the proposed Deep and Wide neural network architecture to support MI discrimination.

**Figure 3 sensors-21-05105-f003:**
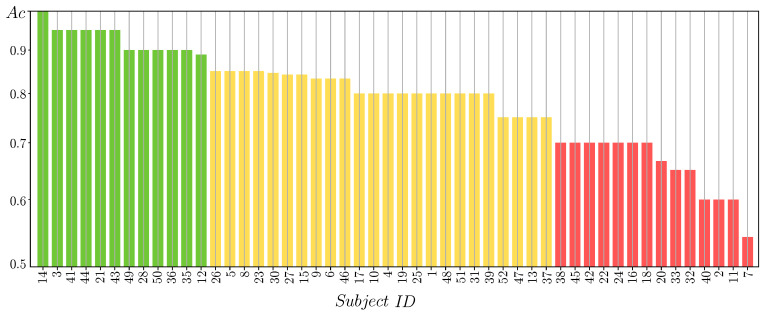
Partitions of individuals clustered by the MLP-based accuracy. Each subject performance is painted by this estimated BCI inefficiency partition: Group *I* (green), Group *II* (yellow), and Group *III* (red).

**Figure 4 sensors-21-05105-f004:**
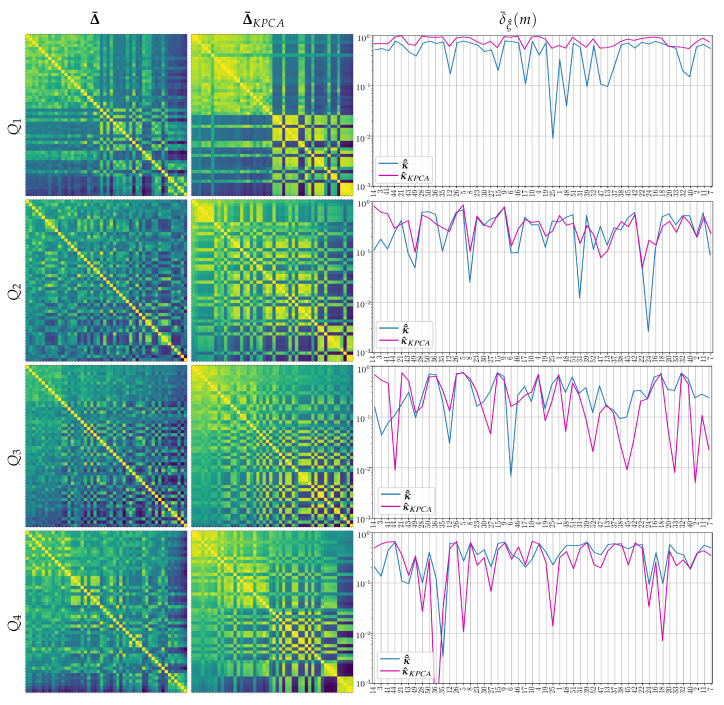
Similarity matrix performed by the tensor product and computed domain marginal values ${\bar{\delta }}_{\hat{\xi }}\left(m\right)$. The subjects are ranked in decreasing order of accuracy.

**Figure 5 sensors-21-05105-f005:**
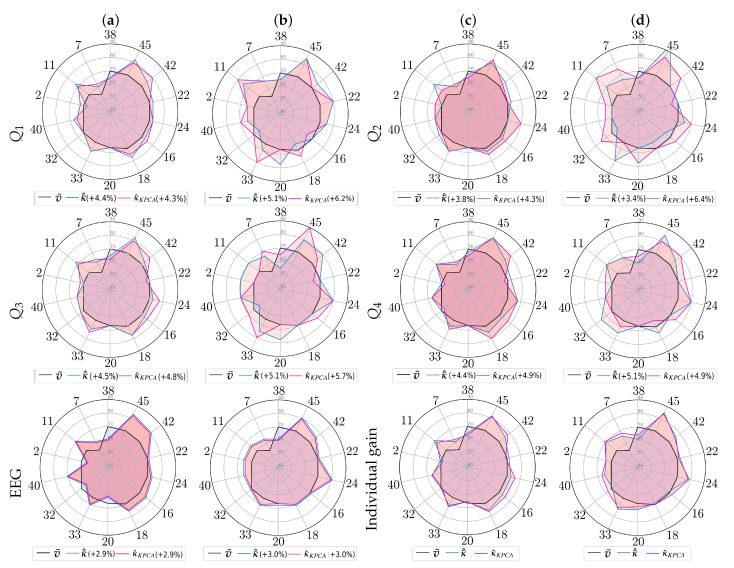
Achieved accuracy by validated strategies of selecting source subjects from Group I. (**a**,**c**) *Single source*-*single-target*, (**b**,**d**) *Multiple sources-single-target*. Individual gain reports the average accuracy per subject of questionnaire data $Q_{i}$ and EEG.

**Figure 6 sensors-21-05105-f006:**
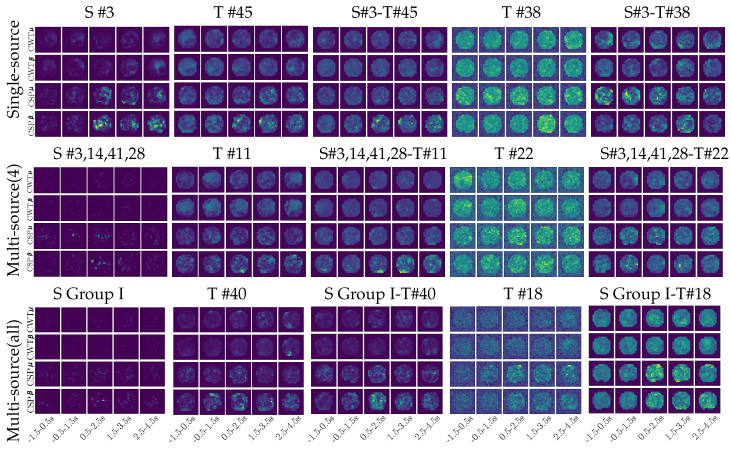
Topographic maps of representative subjects with and without transfer learning using just feature map information, presenting the learned weights with meaningful activity reconstructed within both bandwidths (β and μ) across the whole signal length, $N_{\tau }$.

**Table 1 sensors-21-05105-t001:** Detailed *Deep and Wide* architecture of transfer learning. Layer FC8 accomplishes the regularization procedure using the *Elastic-Net* configuration, while layers FC8 and OU10 apply a kernel constraint adjusted to *max_norm(1.)*. Notation $O=RN_{\Delta }N_{\tau }$, $N_{\Delta }$ denotes the number of filter banks, $P^{\prime }$—the number of hidden units (neurons), *C*—the number of classes, and $I_{L}$ stands for the amount of kernel filters at layer *L*. Notation ||·|| stands for the concatenation operator.

*Layer*	*Assignment*	*Output Dimension*	*Activation*	*Mode*
IN1	Input	||40 × 40||		
CN2	Convolution	||40 × 40 × 2||	ReLu	*Padding* = SAME
				*Size* = 3 × 3
				*Stride* = 1 × 1
BN3	Batch-normalization	||40 × 40 × 2||		
MP4	Max-pooling	||20 × 20 × 2||		*Size* = 2 × 2
				*Stride* = 1 × 1
CT5	Concatenation	$||20\;\times \;20\;\times \;O\cdot I_{L}\left|\right|$		
FL6	Flatten	$20\cdot 20\cdot O\cdot I_{L}$		
BN7	Batch-normalization	$20\cdot 20\cdot O\cdot I_{L}$		
FC8	Fully-connected	$\left|\right|P^{\prime }\;\times \;1\left|\right|$	ReLu	Elastic-Net
				*max_norm(1.)*
BN9	Batch-normalization	$\left|\right|P^{\prime }\;\times \;1\left|\right|$		
OU10	Output	||C × 1||	Softmax	*max_norm(1.)*

**Table 2 sensors-21-05105-t002:** Comparison of bi-class accuracy achieved by state-of-the-art approaches in *GigaScience*. The best value is marked in bold. Notation * denotes Deep and Wide framework results with transfer learning (TL). CSP + FLDA: Common spatial patterns and Fisher linear discriminant analysis, LSTM + Optical: Long-short term memory network and optical predictor, SFCSP: Sparse filter-bank CSP, DCJNN: Deep CSP neural network with joint distribution adaptation, MINE+EEGnet: Mutual information neural estimation, MSNN: Multi-scale Neural Network.

*Approach*	$A_{c}$	*Interpretability*
CSP + FLDA [47]	67.60	–
LSTM + Optical [48]	68.2 ± 9.0	–
SFBCSP [49]	72.60	–
DCJNN [50]	76.50	✓
MINE + EEGnet [51]	76.6 ± 12.48	✓
MSNN [46]	81.0 ± 12.00	✓
Proposal	79.5 ± 10.80	✓
Proposal + TL *	**82.6 ± 8.40**	✓

**Table 3 sensors-21-05105-t003:** Computational time experiments. The achieved training time (average) per fold and epoch is presented.

*Approach*	*Time per Fold*	*Time per Training Epoch*
Proposal (Single-source)	∼984 s	<1 s
Proposal (Multi-source (4))	∼1663 s	<1 s
Proposal (Multi-source (all))	∼3176 s	∼1 s
Proposal + TL	∼341 s	<1 s

## Data Availability

The databases used in this study are public and can be found at the following links: *GigaScience*: http://gigadb.org/dataset/100295, accessed on 10 March 2021.

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
