# Peer review of "Deep and Wide Transfer Learning with Kernel Matching for Pooling Data from Electroencephalography and Psychological Questionnaires"

_sensors, 2021, doi:10.3390/s21155105_

Round 1
Reviewer 1 Report
The following things are not clear to me (it is very hard to understand this research paper):
- network structure with specific parameters (a figure followed by Table 1 would be perfect);
- how to implement real-time/online MI classification when combining psychological questionnaire;
- how to combine the psychological questionnaire data with EEG;
- sizes for the training and test set; how did you divide the dataset;
- detailed performance including training and test time for each subject with/without psychological inputs;
- how to overcome the long training using the proposed method;
- typo: "accessde";
Reviewer 2 Report
This paper presents a transfer learning approach for EEG Motor Imagery. Although it has merit, several points can be considered, such as:
- Some recent papers are missing in the Introduction and in the comparisons with related works, such as:
1) Kaishuo Zhang, Neethu Robinson, Seong-Whan Lee, Cuntai Guan, Adaptive transfer learning for EEG motor imagery classification with deep Convolutional Neural Network, Neural Networks, Volume 136, 2021, Pages 1-10, ISSN 0893-6080, https://doi.org/10.1016/j.neunet.2020.12.013.
2) X. Wei, P. Ortega and A. A. Faisal, "Inter-subject Deep Transfer Learning for Motor Imagery EEG Decoding," 2021 10th International IEEE/EMBS Conference on Neural Engineering (NER), 2021, pp. 21-24, doi: 10.1109/NER49283.2021.9441085.
3) Piyush Kant, Shahedul Haque Laskar, Jupitara Hazarika, Rupesh Mahamune, CWT Based Transfer Learning for Motor Imagery Classification for Brain computer Interfaces, Journal of Neuroscience Methods, Volume 345, 2020, 108886, ISSN 0165-0270, https://doi.org/10.1016/j.jneumeth.2020.108886.
4) Minmin Zheng, Banghua Yang, Shouwei Gao, Xia Meng, Spatio-time-frequency joint sparse optimization with transfer learning in motor imagery-based brain-computer interface system, Biomedical Signal Processing and Control, Volume 68, 2021, 102702, ISSN 1746-8094, https://doi.org/10.1016/j.bspc.2021.102702.
The authors should update the related works section and include recent comparisons in section 4 (even for different datasets).
- The Loss Function in 2b should be better detailed.
- All the terms must be defined in Section 2. Some are missing (e.g., P).
- The notation in Section 2 is highly abstract (maybe more than necessary for the model) for reproducibility. The authors should make the codes publicly available for future comparisons.
- Increase font size in figures 3, 4, and 5.
- Results are comparable with the MSNN method in Table 2. The authors should also discuss computational complexity aspects to emphasize the original contribution.
Round 2
Reviewer 1 Report
Thank you for this clarification, I have no further questions, only missing a period on page 11, line 350.
Reviewer 2 Report
All the suggestion were applied in this new revised version.